# FORTIFIED NETWORKS: IMPROVING THE ROBUSTNESS OF DEEP NETWORKS BY MODELING THE MANIFOLD OF HIDDEN REPRESENTATIONS

## ABSTRACT

A known weakness of deep networks is a failure to perform well when evaluated on data which differ from the training distribution, even if these differences are very small, as is the case with adversarial examples. We propose *Fortified Networks*, a simple extension of existing networks, which "fortifies" the hidden layers in a deep network by identifying when the hidden states are off of the data manifold, and maps these hidden states back to parts of the data manifold where the network performs well. Our principal contribution is to show that fortifying these hidden states improves the robustness of deep networks and our experiments (i) demonstrate improved robustness to standard adversarial attacks in both black-box and white-box threat models; (ii) suggest that our improvements are not primarily due to the problem of deceptively good results due to degraded quality in the gradient signal (the gradient masking problem) and (iii) show the advantage of doing this fortification in the hidden layers instead of the input space. We demonstrate improvements in adversarial robustness on three datasets (MNIST, Fashion MNIST, CIFAR10), across several attack parameters, both white-box and black-box settings, and the most widely studied attacks (FGSM, PGD, Carlini-Wagner). We show that these improvements are achieved across a wide variety of hyperparameters.

## 1 INTRODUCTION

The success of deep neural networks across a variety of tasks has also driven applications in domains where reliability and security are critical, including self-driving cars (Bojarski et al., 2016), health care, face recognition (Sharif et al., 2017), and the detection of malware (LeCun et al., 2015). Security concerns arise when an agent using such a system could benefit from the system performing poorly. Reliability concerns come about when the distribution of input data seen during training can differ from the distribution on which the model is evaluated.

*Adversarial examples* (Goodfellow et al., 2014) result from attacks on neural network models, applying small perturbations to the inputs that change the predicted class. Such perturbations can be small enough to be unnoticeable to the naked eye. It has been shown that gradient-based methods allow one to find modifications of the input that often change the predicted class (Szegedy et al., 2013; Goodfellow et al., 2014). More recent work demonstrated that it is possible to create modifications such that even when captured through a camera, they change the predicted class with high probability (Brown et al., 2017).

Some of the most prominent classes of defenses against adversarial examples include feature squeezing (Xu et al., 2017), adapted encoding of the input (Jacob Buckman, 2018), and distillation-related approaches (Papernot et al., 2015). Existing defenses provide some robustness but most are not easy to deploy. In addition, many have been shown to be providing the illusion of defense by lowering the quality of the gradient signal, without actually providing improved robustness (Athalye et al., 2018). Still others require training a generative model directly in the visible space, which is still difficult today even on relatively simple datasets.

Our work differs from the approaches using generative models in the input space in that we instead employ this robustification on the distribution of the learned hidden representations, which makes the

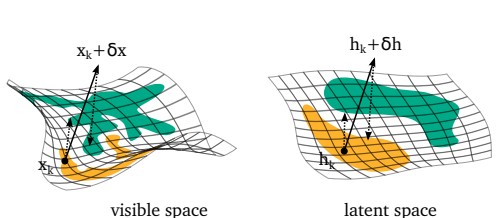 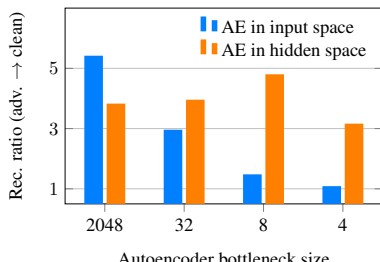

Figure 1: An investigation showing why fortified networks improve robustness. The illustrations on the **left** show how the deeper layers of a network can have a simpler manifold and statistical structure as compared to in the visible space. The plot on the **right** shows direct experimental evidence for this hypothesis: we added fortified layers with different capacities to MLPs trained on MNIST, and display the value of the total reconstruction errors for adversarial examples divided by the total reconstruction errors for clean examples. A high value indicates success at detecting adversarial examples. Our results support the central motivation for fortified networks: that off-manifold points can much more easily be detected in the hidden space (as seen by the relatively constant ratio for the autoencoder in hidden space) and are much harder to detect in the input space (as seen by this ratio rapidly falling to zero as the input-space autoencoder's capacity is reduced).

identification of off-manifold examples easier. We do this by training denoising autoencoders on top of the hidden layers of the original network. We call this method Fortified Networks.

We demonstrate that Fortified Networks (i) can be generically added into an existing network; (ii) robustify the network against adversarial attacks and (iii) provide a reliable signal of the existence of input data that do not lie on the manifold on which it the network trained.

In the sections that follow, we discuss the intuition behind the fortification of hidden layers and lay out some of the method's salient properties. Furthermore, we evaluate our proposed approach on MNIST, Fashion-MNIST, CIFAR10 datasets against whitebox and blackbox attacks.

## 2 BACKGROUND

**The Empirical Risk Minimization Framework**    Let us consider a standard classification task with an underlying data distribution $\mathcal{D}$ over pairs of examples $x \in \mathbb{R}^d$ and corresponding labels $y \in [k]$. We also assume that we are given a suitable loss function $L(\theta, x, y)$, for instance the cross-entropy loss. As usual, $\theta \in \mathbb{R}^p$ is the set of model parameters. Our goal then is to find model parameters $\theta$ that minimize the risk $\mathbb{E}_{(x,y)\sim\mathcal{D}}[L(x, y, \theta)]$. This expectation cannot be computed, therefore a common approach is to to minimize the empirical risk $1/N \sum_D L(x, y, \theta)$ taking into account only the examples in a given dataset $D$.

**Adversarial Attacks and Robustness**    While the empirical risk minimization framework has been very successful and often leads to excellent generalization, it has the significant limitation that it doesn't guarantee robustness, and more specifically performance on examples off the data manifold. Madry et al. (2017) proposed an optimization view of adversarial robustness, in which the adversarial robustness of a model is defined as a min-max problem,

$$\min_{\theta} \rho(\theta), \quad \text{where} \quad \rho(\theta) = \mathbb{E}_{(x,y)\sim\mathcal{D}}\left[\max_{\delta\in\mathcal{S}} L(\theta, x + \delta, y)\right], \tag{1}$$

where $\mathcal{S}$ denotes the set of all points within a sphere of radius $\varepsilon$, which is task-specific. Larger $\varepsilon$ values correspond to stronger attacks but which may be more visually apparent.

**Denoising Autoencoders**    *Denoising autoencoders* (DAEs) are neural networks which take a noisy version of an input (for example, an image) and are trained to predict the noiseless version of that input. This approach has been widely used for feature learning and generative modeling in deep learning (Bengio et al., 2013a). More formally, denoising autoencoders are trained to minimize a reconstruction error or negative log-likelihood of generating the clean input. For example, with

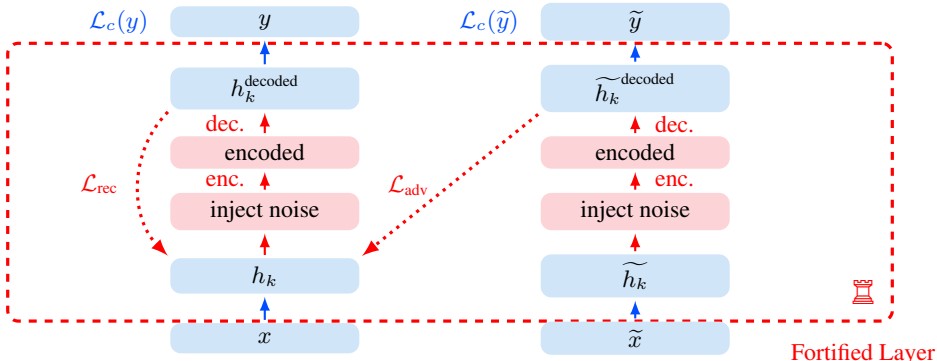

Figure 2: Diagram illustrating a one-layer fortified network. A network is evaluated with a data sample $x$ and its corresponding adversarial example $\widetilde{x}$. Hidden units $h_k$ and $\widetilde{h_k}$ are corrupted with noise, encoded with the encoder *enc.*, and decoded with the decoder *dec.* The autoencoder (denoted by the red color) is trained to reconstruct the hidden unit $h_k$ that corresponds to the clean input. Dotted lines are two reconstruction costs: for a benign ($\mathcal{L}_{rec}$) and adversarial examples ($\mathcal{L}_{adv}$).

Gaussian log-likelihood of the clean input given the corrupted input, $r_\theta$ the learned denoising function, $C$ a corruption function with Gaussian noise of variance $\sigma^2$, the reconstruction loss is

$$\widehat{\mathcal{L}} = \frac{1}{N} \sum_{n=1}^{N} \left( \left\| r(C_\sigma(x^{(n)})) - x^{(n)} \right\|_2^2 \right). \tag{2}$$

Alain et al. (2012) demonstrated that with this loss function, an optimally trained denoising autoencoder's reconstruction vector is proportional to the gradient of the log-density:

$$\frac{r_\sigma(x) - x}{\sigma^2} \rightarrow \frac{\partial \log p(x)}{\partial x} \quad \text{as} \quad \sigma \rightarrow 0. \tag{3}$$

This theory establishes that the reconstruction vectors from a well-trained denoising autoencoder form a vector field which points in the direction of the data manifold. However, Alain et al. (2012) showed that this may not hold for points which are distant from the manifold, as these points are rarely sampled during training. In practice, denoising autoencoders are not just trained with tiny noise but also with large noises, which blurs the data distribution as seen by the learner but makes the network learn a useful vector field even far from the data.

## 3    FORTIFIED NETWORKS

We propose the use of DAEs inserted at crucial points between layers of the original neural network in order to denoise the transformed data points which may lie away from the original data manifold. Intuitively, the method aims to regularize the hidden representations by keeping the activations on the surface of the corresponding projected data manifold through the application of a DAE trained on the hidden representations (on the original clean data). We argue that applying the DAEs on the hidden layers—as opposed to the raw input signal—facilitates learning, while providing a stronger protection from adversarial attacks. As illustrated in Figure 1, we hypothesize that more abstract representations associated with deeper networks are easier to denoise because the transformed data manifolds are flatter. The flattening of data manifolds in the deeper layers of a neural network was first noted experimentally by Bengio et al. (2013b). We provide experimental support for these claims in Section 4.

**Layer fortification** ♖    Our method works by substituting a hidden layer $h_k$ with a denoised version. We feed the signal $h_k$ through the encoder network, $E_k$, and decoder network, $D_k$, of a DAE for layer $k$, which yields the denoised version, $h_k^{decoded}$:

$$h_k^{decoded} = D_k(E_k(h_k + n_k)), \tag{4}$$

where $n_k$ is white Gaussian noise of variance $\sigma^2$ and appropriate shape. We call the resulting layer, a *fortified layer* and the resulting network the *fortified network* corresponding to the original network.

- **Reconstruction loss.** For a mini-batch of $N$ clean examples, $x^{(1)}, \ldots, x^{(N)}$, each hidden layer $h_k^{(1)}, \ldots, h_k^{(N)}$ is fed into a DAE loss, similar to equation 2:

$$\mathcal{L}_{rec,k} = \frac{1}{N} \sum_{n=1}^{N} \left\| D_k \left( E_k \left( h_k^{(n)} + n_k \right) \right) - h_k^{(n)} \right\|_2^2 .$$

- **Adversarial loss.** We use some adversarial training method to produce the perturbed version of the mini-batch, $\widetilde{x}^{(1)}, \ldots, \widetilde{x}^{(N)}$, where $\widetilde{x}^{(i)}$ is a small perturbation of $x^{(i)}$ which is designed to make the network produce the wrong answer. The corresponding hidden layer $\widetilde{h}_k^{(1)}, \ldots, \widetilde{h}_k^{(N)}$ (using the perturbed rather than original input) is fed into a similar DAE loss:

$$\mathcal{L}_{adv,k} = \frac{1}{N} \sum_{n=1}^{N} \left\| D_k \left( E_k \left( \widetilde{h}_k^{(n)} + n_k \right) \right) - h_k^{(n)} \right\|_2^2 ,$$

where we note that the target reconstruction for denoising is the clean version of the hidden layer, without noise and without adversarial perturbation.

For training purposes, we treat the DAEs as part of the fortified network, backpropagate through and train all weights jointly. Aside from the original classification loss, $\mathcal{L}_c$, we also include the classification loss from the adversarial objective, $\mathcal{L}_c(\widetilde{y})$, and we introduce a dual objective for the DAEs.

To build a fortified network, we can apply this fortification process to some or all the layers. The final objective used for training the fortified network includes the classification loss and all reconstruction and adversarial losses:

$$\mathcal{L} = \mathcal{L}_c(y) + \mathcal{L}_c(\widetilde{y}) + \lambda_{rec} \sum_k \mathcal{L}_{rec,k} + \lambda_{adv} \sum_k \mathcal{L}_{adv,k},$$

where $\lambda_{rec} > 0$ and $\lambda_{adv} > 0$ tune the strength of the DAE terms. This kind of training process allows for the production of hidden representations robust to small perturbations, and in particular, to adversarial attacks.

**Off-manifold signaling** The reconstruction losses act as a reliable signal for detecting off-manifold examples (cf. Section 4). This is a particularly useful property in practice: not only can we provide more robust classification results, we can also sense and suggest to the analyst or system when the original example is either adversarial or from a significantly different distribution.

**Motivation for when and where to use fortified layers** We have discussed advantages to placing fortified layers in the hidden states instead of the input space (with further discussion in section 6,) but the question of where exactly fortified layers need to be placed remains unanswered. Is it just the final hidden layer? Is it every hidden layer? We outline two important considerations regarding this issue. First, in the higher-level hidden layers, it is much easier for the network to identify points which are off of the manifold or close to the margin. This is directly experimentally demonstrated in fig. 1. Secondly, the higher level hidden layers may already look like points that are not adversarial due to the effect of the adversarial perturbations in the earlier layers. While we are not aware of any formal study of this phenomenon, it is clearly possible (imagine for example a fortified layer on the output from the softmax, which could only identify unnatural combinations of class probabilities.) Given these opposing objectives, we argue for the inclusion of multiple fortified layers across the network.

## 4 EXPERIMENTS

### 4.1 ATTACKS

We evaluated the performance of our model as a defense against adversarial attacks. We focused on two of the most popular and well-studied attacks. Firstly, we consider the Fast Gradient Sign

Method (FGSM, Goodfellow et al., 2014) which is popular as it only requires a single step and can still be effective against many networks. Secondly, we consider the projected gradient descent attack (Kurakin et al., 2016) which is slower than FGSM as it requires many iterations, but has been shown to be a much stronger attack (Madry et al., 2017).

Additionally, we consider both white-box attacks (where the attackers knows the model) and black-box attacks (where they don't, but they have access to the training set.)

**Fast Gradient Sign Method**. The Fast Gradient Sign Method (FGSM) Goodfellow et al. (2014) is a simple one-step attack that produces $\ell_\infty$-bounded adversaries via the following gradient based perturbation.

$$\widetilde{x} = x + \varepsilon \operatorname{sgn}(\nabla_x L(\theta, x, y)). \tag{5}$$

**Projected Gradient Descent**. The projected gradient descent attack (Madry et al., 2017), sometimes referred to as FGSM$^k$, is a multi-step extension of the FGSM attack characterized as follows:

$$x^{t+1} = \Pi_{x+\mathcal{S}} \left( x^t + \alpha \operatorname{sgn}(\nabla_x L(\theta, x, y)) \right) \tag{6}$$

initialized with $x^0$ as the clean input $x$ and with the corrupted input $\widetilde{x}$ as the last step in the sequence. $\Pi$ refers to the projection operator, which in this context means projecting the adversarial example back onto the region within an $\varepsilon$ radius of the original data point, after each step in the adversarial attack.

Finally we considered the Carlini-Wagner L2 attack (Carlini & Wagner, 2016) which consists of joint optimization of loss maximization and minimizing the distance of the adversarial example to the original example.

## 4.2 The Gradient Masking and Gradient Obfuscation Problem

A significant challenge with evaluating defenses against adversarial attacks is that many attacks rely upon a network's gradient. Methods which reduce the quality of this gradient, either by making it flatter or noisier can lead to methods which lower the effectiveness of gradient-based attacks, but which are not actually robust to adversarial examples (Athalye et al., 2017; Papernot et al., 2016c). This process, which has been referred to as gradient masking or gradient obfuscation, must be analyzed when studying the strength of an adversarial defense.

One method for studying the extent to which an adversarial defense gives deceptively good results as a result of gradient masking relies on the observation that black-box attacks are a strict subset of white-box attacks, so white-box attacks should always be at least as strong as black-box attacks. If a method reports much better defense against white-box attacks, it suggests that the selected white-box attack is underpowered as a result of gradient masking. Another test for gradient masking is to run an iterative search, such as projected gradient descent (PGD) with an unlimited range for a large number of iterations. If such an attack is not completely successful, it indicates that the model's gradients are not an effective method for searching for adversarial images, and that gradient masking is occurring. Still another test is to confirm that iterative attacks with small step sizes always outperform single-step attacks with larger step sizes (such as FGSM). If this is not the case, it may suggest that the iterative attack becomes stuck in regions where optimization using gradients is poor due to gradient masking.

Additionally, (Athalye et al., 2018) discussed the Backward Pass Differentiable Approximation (BPDA) attack to cover cases where a defense employs a transformation which is clearly non-differentiable or reduces the quality of the gradients. Because we pass gradients through the fortified layers in the normal training of our network, it is unlikely that the quality of these gradients is significantly deteriorated, and there isn't a reason to expect that they would be because the fortified layers are relatively shallow and use normal activation functions (i.e. no non-differentiable functions.). Additionally we ran additional experiments using the identity function version of BPDA on Fashion MNIST FGSM ($\varepsilon = 0.3$) with standard deviation across five trials. Our adversarial training baseline achieved an accuracy of 84.64±0.48 on this task, fortified networks with the normal attack achieved an accuracy of 89.50±0.45, and fortified networks with the BPDA version of the attack (treating the autoencoder as an identity) achieved an accuracy of 89.88±0.30, which corresponds to a weaker attack.

## 5 RESULTS

For details about the specifics of our model architectures and hyperparameters we refer readers to the Appendix. With all experiments, we use the same attacks (with identical parameters) at training and test time to generate adversarial examples. An important point to note here is that all of the autoencoders in our fortified layers used a single hidden layer with tied weights.

Table 1: Results on white-box MNIST and CIFAR with FGSM . The ResNet model provided in (Papernot et al., 2016a) was used in the CIFAR FGSM experiments.

MNIST

| | | |
|---|---|---|
| FGSM $\varepsilon = 0.3$ | Adv. Train (Madry) | 95.60 |
| | Adv. Train (ours) | 96.36 |
| | Adv. Train No-Rec (ours) | 96.47 |
| | Fortified Net, no $\mathcal{L}_{adv}$ (ours) | 96.46 |
| | Fortified Network (ours)* | **97.97** |
| CW L2 | Adv. Train | 46.16 |
| | Fortified Network | **60.58** |
| PGD $\varepsilon = 0.1$ | Baseline Adv. Train | 96.98 |
| | Fortified Network (ours) | **98.09** |

CIFAR-10

| | | |
|---|---|---|
| FGSM $\varepsilon = 0.3$ | Baseline Adv. Train | 79.57 |
| | Fortified Networks (ours) | **80.47** |
| FGSM $\varepsilon = 0.03$ | Baseline Adv. Train (ours) | 79.34 |
| | *Fortified Networks (ours)* | |
| | Autoencoder (fortify input) | 79.77 |
| | Autoencoder in hidden space | **81.80** |

We also performed an analytical experiment on an RNN language model to study if fortified networks can detect when it's being given outputs from its own model (sampling mode) when trained using ground truth input sequences. To this end we train a language model on the standard Text8 dataset, which is derived from Wikipedia articles. We trained a single-layer LSTM with 1000 units at the character-level, and included fortified layers between the hidden states and the output on each time step.

With 50 sampling steps, the fortified layers had a reconstruction error on average 103% of the teacher forcing reconstruction error. With 180 sampling steps, this valued increased to 112%. With 300 sampling steps this increased even further to 134%. This is clear evidence that the outputs move off of the manifold with more sampling steps, and that this is effectively measured by fortified networks.

**Analysis of Hyperparameters** We ran with many hyperparameters for the fortified layers to demonstrate the generality of the improvement given by fortified networks. We ran FGSM on Fashion-MNIST ($\varepsilon$=0.3) while varying the amount of noise injected and the weighting on the reconstruction loss. We also varied the amount of noise and choice of losses in the ablation experiment in Table 7. We achieved consistent improvement across a variety of settings but found the largest improvement when using both the adversarial and clean reconstruction losses and a small amount of noise.

Thus, we see that a consistent improvement is achieved when the weighting on the reconstruction part of the loss and the amount of noise injected are varied over several orders of magnitude.

## 6 RELATED WORK

**Using Generative Models as a Defense**. The observation that adversarial examples often consist of points off of the data manifold and that deep networks may not generalize well to these points motivated (Gu & Rigazio, 2014; Ilyas et al., 2017; Samangouei et al., 2018; Liao et al., 2017) to consider the use of the generative models as a defense against adversarial attacks. Ilyas et al. (2017); Gilmer et al. (2018) also showed the existence of adversarial examples which lie on the data manifold, and (Ilyas et al., 2017) showed that training against adversarial examples forced to lie on the manifold is an effective defense. Our method shares a closely related motivation to these prior works, with a key difference being that we propose to consider the manifold in the space of learned representations,

---

[1]The MNIST experiment with FGSM was also run for $\varepsilon = 0.1$ to compare directly with (Erraqabi et al., 2018). We obtain **98.34%** on adversarial examples compared to their 96.10%.

Table 2: CIFAR-10 PGD Results with (non-resnet) CNNs. In these experiment we used a fortified block (single convolutional autoencoder) following each convolutional layer. Both experiments were run for 200 epochs and with all hyperparameters and architecture kept the same with the exception of the fortified layer being added. We considered different types of baselines: 'Baseline - no new layers' means we simply removed the fortified block. 'Baseline - extra layers' means that we added extra layers to match the capacity of the fortified layers, but only gave half of these extra layers activations as the fortified block has two layers but only one activation. 'Baseline - extra activations' means that we added an activation following each layer, giving more activations in total than the Fortified Network.

| Method | Attack Type | PGD Steps | Attack Epsilon | PGD Accuracy |
|---|---|---|---|---|
| Baseline - extra activations | Normal | 7 | 0.03 | 38.1 |
| Fortified Nets | Normal | 7 | 0.03 | 43.3 |
| Baseline - no new layers | Normal | 7 | 0.03 | 33.0 |
| Baseline - no new layers | Normal | 50 | 0.03 | 31.6 |
| Baseline - no new layers | Normal | 200 | 0.03 | 31.4 |
| Baseline - extra layers | Normal | 7 | 0.03 | 34.2 |
| Baseline - extra layers | Normal | 50 | 0.03 | 32.5 |
| Baseline - extra layers | Normal | 200 | 0.03 | 32.2 |
| Fortified Networks | Normal | 7 | 0.03 | 45.0 |
| Fortified Networks | Normal | 50 | 0.03 | 42.1 |
| Fortified Networks | Normal | 200 | 0.03 | 41.5 |
| Baseline - extra layers | Normal | 100 | 0.03 | 35.3 |
| Baseline - extra layers | Normal | 100 | 0.04 | 24.8 |
| Baseline - extra layers | Normal | 100 | 0.06 | 14.3 |
| Baseline - extra layers | Normal | 100 | 0.08 | 12.0 |
| Baseline - extra layers | Normal | 100 | 0.1 | 11.7 |
| Baseline - extra layers | Normal | 100 | 0.2 | 10.2 |
| Baseline - extra layers | Normal | 100 | 0.3 | 8.4 |
| Fortified Networks | Normal | 100 | 0.03 | 39.2 |
| Fortified Networks | Normal | 100 | 0.04 | 28.0 |
| Fortified Networks | Normal | 100 | 0.06 | 15.6 |
| Fortified Networks | Normal | 100 | 0.08 | 13.0 |
| Fortified Networks | Normal | 100 | 0.1 | 12.9 |
| Fortified Networks | Normal | 100 | 0.2 | 11.3 |
| Fortified Networks | Normal | 100 | 0.3 | 9.6 |
| Baseline - extra layers | Normal | 100 | 0.03 | 33.4 |
| Fortified Networks | Normal | 100 | 0.03 | 40.1 |
| Fortified Networks | Noiseless Attack | 100 | 0.03 | 38.2 |
| Fortified Networks | BPDA, Skip-AE | 100 | 0.03 | 67.1 |
| Fortified Networks | Normal | 7 | 0.03 | 43.3 |
| Baseline - extra layers | Normal | 7 | 0.03 | 38.1 |
| Baseline - extra layers | ALP-Like | 7 | 0.03 | 34.2 |
| Fortified Networks | Normal | 100 | 0.03 | 39.20 |
| Baseline - extra layers | Normal | 100 | 0.03 | 35.3 |
| Baseline - extra layers | ALP-Like | 100 | 0.03 | 32.2 |

instead of considering the manifold directly in the visible space. One motivation for this is that the learned representations have a simpler statistical structure (Bengio et al., 2012), which makes the task of modeling this manifold and detecting unnatural points much simpler. Learning the distribution directly in the visible space is still very difficult (even state of the art models fall short of real data on metrics like Inception Score) and requires a high capacity model. Additionally, working in the space of learned representations allows for the use of a relatively simple generative model, in our case a small denoising autoencoder.

Ilyas et al. (2017) proposed to work around these challenges from working in the visible space by using the Deep Image Prior instead of an actual generative model. While this has the advantage of

Table 3: CIFAR-10 PGD Results with ResNets. In this experiment we used a single fortified layer following the 2nd resblock, and the baseline consists of the same network but with the fortified layer removed. Both experiments were run for 200 epochs and with all hyperparameters and architecture kept the same with the exception of the fortified layer being added.

| Model | Method | PGD Accuracy (20 steps) | Clean Test Accuracy |
|---|---|---|---|
| PreActResNet18 | Baseline | 37.87 | 84.93 |
| PreActResNet18 | Fortified Networks | 39.20 | 84.84 |
| WideResNet28-10 | Baseline | 43.28 | 87.42 |
| WideResNet28-10 | Fortified Networks | 44.06 | 87.40 |

Table 4: Accuracies against white-box attacks on Fashion MNIST. For PGD we used $\varepsilon = 0.1$ and for FGSM we used $\varepsilon = 0.1$ and $\varepsilon = 0.3$[1]. Compared with DefenseGAN (Samangouei et al., 2018).

| Model | FGSM ($\varepsilon = 0.1$) | FGSM ($\varepsilon = 0.3$) | PGD ($\varepsilon = 0.1$) |
|---|---|---|---|
| DefenseGAN | n/a | 89.60 | n/a |
| Baseline Adv. Train - Conv,ReLU | 86.14 | 90.66 | 77.49 |
| Baseline Adv. Train - Conv,LReLU | 89.10 | 88.8 | 77.90 |
| Fortified Nets - Conv (ours) | **89.86** | **91.31** | **79.54** |

Table 5: **Left**: Accuracies against blackbox MNIST attacks with adversarial training (FGSM). Reporting 50/50 results compared to previous works (Samangouei et al., 2018, PS). The test error on clean examples is in parenthesis. **Right**: We ran a fortified network on Fashion-MNIST using adversarial training with PGD for a variety of $\varepsilon$ values, each for 5 epochs. The motivation behind this experiment, suggested by Athalye et al. (2018) is confirming if unbounded ($\varepsilon = 1$) adversarial attacks are able to succeed. A defense which succeeds primarily by masking or obfuscating the gradients would fail to bring the accuracy to zero even with an unbounded attack. As can be seen, unbounded attacks against Fortified Networks succeed when given a sufficiently large $\varepsilon$, which is evidence against gradient masking.

| | |
|---|---|
| DefenseGAN fc→conv (PS) | 92.21 (n/a) |
| DefenseGAN conv→conv (PS) | 93.12 (n/a) |
| Adv. Train fc→conv (PS) | 96.68 (n/a) |
| Adv. Train conv→conv (PS) | 96.54 (n/a) |
| *Our Approaches* | |
| Baseline Adv. Train | 93.83 (98.95) |
| Fortified Net w/o $\mathcal{L}_{adv}$, $\mathcal{L}_{rec}$ | 96.98 (99.17) |
| Fortified Network | **97.82** (98.93) |

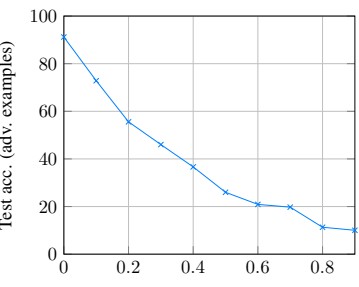

$\varepsilon$ for PGD search (1 means unbounded)

being a model that doesn't require a special training procedure (as deep image prior is a separate optimization process for each example) it may be limited in the types of adversarial attacks that it's resistant to, and it would provide no defense against adversarial attacks which are in the range of a convolutional network, which have been shown to exist (Xiao et al., 2018).

Another key difference between our work and (Ilyas et al., 2017; Samangouei et al., 2018) is that both DefenseGAN and the Invert-and-Classify approach use an iterative search procedure at inference time to map observed data points onto nearby points on the range of the generator. On the other hand, our approach uses small denoising autoencoders that are used in the same way (i.e. a simple forward application) during both training and testing. The use of such an iterative procedure presents challenges for evaluation, as it is possible for gradients to vanish while doing backpropagation through such a procedure, which may lead to an overestimate in the strength of the defense due to the gradient masking problem (Papernot et al., 2016b; Athalye et al., 2018). One indicator of the gradient masking problem is black-box attacks outperforming white-box attacks, which is an indicator of

Table 6: Hyperparameter combinations (Fashion-MNIST; FGSM; $\varepsilon = 0.3$)

| Reconstruction Loss Weight | n/a | 0.01 | 0.1 | 1.0 | 1.0 | 1.0 | 1.0 |
|---|---|---|---|---|---|---|---|
| Autoencoder Noise ($N(0, \sigma)$) | n/a | 0.01 | 0.01 | 0.01 | 0.001 | 0.01 | 0.1 |
| Accuracy | 88.00 | 89.70 | 90.01 | 91.00 | 90.78 | 91.00 | 91.31 |

Table 7: Fashion MNIST, PGD ($\varepsilon = 0.1$), 40 attack iterations, 50 epochs; experiments based on a 2 hidden layer MLP with 512 units per layer and leaky relu activation. $\pm$ standard deviation reported over last 5 epochs. All setups improve over the baseline (72.36%).

| Autoencoder after layer | | | Noise | $L_{\text{rec}}$ | $L_{\text{adv}}$ | Test acc. |
| Input | 1st | 2nd | | | | (%) |
|---|---|---|---|---|---|---|
| - | - | - | n/a | n/a | n/a | $72.36 \pm 0.42$ |
| - | - | ✓ | 0.1 | ✓ | ✓ | $73.44 \pm 0.46$ |
| - | - | ✓ | 0.1 | - | ✓ | $73.69 \pm 0.58$ |
| - | - | ✓ | 0.0 | ✓ | ✓ | $73.39 \pm 0.52$ |
| - | - | ✓ | 0.01 | ✓ | - | $73.12 \pm 0.52$ |
| - | - | ✓ | 0.01 | - | ✓ | $73.46 \pm 0.41$ |
| - | ✓ | ✓ | 0.01 | ✓ | ✓ | $73.64 \pm 0.29$ |
| - | ✓ | ✓ | 0.01 | - | ✓ | $73.46 \pm 0.46$ |
| ✓ | ✓ | ✓ | 0.01 | ✓ | ✓ | $73.78 \pm 0.31$ |
| ✓ | ✓ | ✓ | 0.01 | ✓ | - | $73.36 \pm 0.28$ |
| ✓ | ✓ | ✓ | 0.01 | - | ✓ | $73.27 \pm 0.46$ |

under-powered attacks as black-box attacks are a strict subset of white-box attacks. This indicator of gradient obfuscation was present in the work of Samangouei et al. (2018) where black-box attacks were generally stronger against their defense, but with our method we observe very similar defense quality against black-box and white-box attacks. (Gu & Rigazio, 2014; Liao et al., 2017) both considered using an autoencoder as a pre-processing step in the input space. Interestingly (Liao et al., 2017) used a loss function defined in the space of the hidden states, but still used autoencoders directly in the input space.

**Adversarial Hidden State Matching**. Erraqabi et al. (2018) demonstrate that adversarially matching the hidden layer activations of regular and adversarial examples improves robustness. This work shared the same motivation of using the hidden states to improve robustness, but differed in that they used an adversarial objective and worked in the original hidden states instead of using a generative model (in our case, the DAE in the fortified layers). We present direct experimental comparisons with their work in section 5. (Kannan et al., 2018) proposed a method which involves matching the logit (pre-softmax outputs) values for the original samples with the logit values resulting from adversarial examples.

**Denoising Feature Matching**. Warde-Farley & Bengio (2016) proposed to train a denoising autoencoder in the hidden states of the discriminator in a generative adversarial network. The generator's parameters are then trained to make the reconstruction error of this autoencoder small. This has the effect of encouraging the generator to produce points which are easy for the model to reconstruct, which will include true data points. Both this and Fortified Networks use a learned denoising autoencoder in the hidden states of a network. A major difference is that the denoising feature matching work focused on generative adversarial networks and tried to minimize reconstruction error through a learned generator network, whereas our approach targets the adversarial examples problem. Additionally, our objective encourages the output of the DAE to denoise adversarial examples so as to point back to the hidden state of the original example, which is different from the objective in the denoising feature matching work, which encouraged reconstruction error to be low on states from samples from the generator network.

**Adversarial Spheres**. Gilmer et al. (2018) studied the existence of adversarial examples in the task of classifying between two hollow concentric shells. Intriguingly, they prove and construct adversarial examples which lie on the data manifold (although Ilyas et al. (2017) also looked for such examples experimentally using GANs.) The existence of such on-manifold adversarial examples demonstrates

that a simplified version of our model trained with only $\mathcal{L}_{rec}$ could not protect against all adversarial examples. However, training with $\mathcal{L}_{adv}$ encourages the fortified layers to map back from points which are not only off of the manifold, but also to map back from points which are hard to classify, allowing Fortified Networks to also potentially help with on-manifold adversarial examples as well.

## 7 CONCLUSION

Protecting against adversarial examples could be of paramount importance in mission-critical applications. We have presented Fortified Networks, a simple method for the robustification of existing deep neural networks. Our method is practical, as fortifying an existing network entails introducing DAEs between the hidden layers of the network, which can be automated. Furthermore, the DAE reconstruction error at test time is a reliable signal of distribution shift, which can result in examples unlike those encountered during training. High error can signify either adversarial attacks or significant domain shift; both are important cases for the analyst or system to be aware of. Moreover, fortified networks are efficient: since not every layer needs to be fortified to achieve improvements, fortified networks are an efficient way to improve robustness to adversarial examples. For example, we have shown improvements on ResNets where only two fortified layers are added, and thus the change to the computational cost is very slight. Finally, fortified networks are effective, as they improve results on adversarial defense on three datasets (MNIST, Fashion MNIST, and CIFAR10), across a variety of attack parameters (including the most widely used $\varepsilon$ values), across three widely studied attacks (FGSM, PGD, Carlini-Wagner L2), and in both the black-box and white-box settings.

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

## A    EXPERIMENTAL SETUP

All attacks used in this work were carried out using the Cleverhans  (Papernot et al., 2016a) library.

### A.1    WHITE-BOX ATTACKS

Our convolutional models (Conv, in the tables) have 2 strided convolutional layers with 64 and 128 filters followed by an unstrided conv layer with 128 filters. We use ReLU activations between layers then followed by a single fully connected layer. The convolutional and fully-connected DAEs have a single bottleneck layer with leaky ReLU activations with some ablations presented in the table below.

With white-box PGD attacks, we used only convolutional DAEs at the first and last conv layers with Gaussian noise of $\sigma = 0.01$ whereas with FGSM attacks we used a DAE only at the last fully connected layer. The weight on the reconstruction error $\lambda_{rec}$ and adversarial cost $\lambda_{adv}$ were set to 0.01 in all white-box attack experiments. We used the Adam optimizer with a learning rate of 0.001 to train all models.

The table below lists results a few ablations with different activation functions in the autoencoder

Table 8: More detailed version of table 1, but with more detailed ablation experiments for our method included. Accuracies against white-box MNIST attacks with FGSM, where the model is a convolutional net. We used the standard FGSM attack parameters with an $\varepsilon$ of 0.3 and compare against published adversarial training defenses. We also performed ablations where we considered removing the reconstruction error on adversarial examples $\mathcal{L}_{adv}$ as well as switching the activation function in the fortified layers from leaky relu to tanh, which we found to slightly help in this case.

| Model | FGSM |
|---|---|
| Adv. Train (Madry et al., 2017) | 95.60 |
| Adv. Train (Jacob Buckman, 2018) | 96.17 |
| Adv. Train (ours) | 96.36 |
| Adv. Train No-Rec (ours) | 96.47 |
| Quantized (Jacob Buckman, 2018) | 96.29 |
| One-Hot (Jacob Buckman, 2018) | 96.22 |
| Thermometer (Jacob Buckman, 2018) | 95.84 |
| *Our Approaches* | |
| Fortified Network - Conv, w/o $\mathcal{L}_{adv}$, lrelu | 96.46 |
| Fortified Network - Conv, lrelu | 97.69 |
| Fortified Network - Conv, tanh | **97.97** |

### A.2    BLACK-BOX ATTACKS

Our black-box results are based on a fully-connected substitute model (input-200-200-output), which was subsequently used to attack a fortified convolutional network. The CNN was trained for 50 epochs using adversarial training, and the predictions of the trained CNN were used to train the substitute model. 6 iterations of Jacobian data augmentation were run during training of the substitute, with $\lambda = 0.1$. The test set data holdout for the adversary was fixed to 150 examples. The learning rate was set to 0.003 and the Adam optimizer was used to train both models.

## B    GRADIENT OBFUSCATION

Various validation experiments have been run to show that fortification layers do not merely operate by obfuscating gradients. All results reported in this section are based on ConvNets trained on the CIFAR-10 dataset and attacked with PGD.

| Epsilon | Baseline (extra layers) | Fortified Networks |
|---------|-------------------------|--------------------|
| 0.03 | 35.3 | 39.2 |
| 0.04 | 24.8 | 28.0 |
| 0.06 | 14.3 | 15.6 |
| 0.08 | 12.0 | 13.0 |
| 0.1 | 11.7 | 12.9 |
| 0.2 | 10.2 | 11.3 |
| 0.3 | 8.4 | 9.6 |

Table 9: PGD, attack run for 100 iterations

| Steps | Baseline | Baseline (extra layers) | Fortified Networks |
|-------|----------|-------------------------|--------------------|
| 7 steps | 33.0 | 34.2 | 45.0 |
| 50 steps | 31.6 | 32.5 | 42.1 |
| 200 steps | 31.4 | 32.2 | 41.5 |

Table 10: More attack steps to uncover gradient masking effects.

**Different epsilon values at attack time.** We applied attacks of different $\varepsilon$ values to a network trained with $\varepsilon = 0.03$. Results shown in tab. 9.

**More attack iterations.** We ran attacks for more steps at test time, on a network trained on attacks of 7 steps. Results shown in tab. 10.

