# OpenReview forum: "Fortified Networks: Improving the Robustness of Deep Networks by Modeling the Manifold of Hidden Representations"
_ICLR.cc/2019/Conference_

### Official Review · AnonReviewer3 · 2018-11-03
**good work**

**Rating:** 6
**Confidence:** 3

**Review:**

In this paper, the authors proposed a fortified network model, which is an extension to denoising autoencoder. The extension is to perform the denoising module in the hidden layers instead of input layer. The motivation of this extension is that the denoising part is more effective in the hidden layers. Overall, this extension is quite sensible, and empirical results justify the utility of this extension. The major issue, which was left as an open question in the end of Section 3, is that when and where to use fortified layers. The authors discussed this issue, but did not solve this issue. Nevertheless, I do believe solving this issue requires a sequence of papers. Overall the paper reads very well, but there are a number of minor places to be improved.


(1) a grammar error at "provide a reliable signal of the existence of input data that do not lie on the manifold on which it the network trained."

(2) a grammar error at "This expectation cannot be computed, therefore a common approach is to to minimize the empirical risk"

(3) The sentence "For a mini-batch of N clean examples, x(1), ..., x(N), each hidden layer h(1)_k, ..., h(N)_k is fed into a DAE loss" is a little confusing to me. "h(1)_k, ..., h(N)_k" is only for one hidden layer, rather than "each hidden layer". Right?

---

> ### Author Response · Authors · 2018-11-15
> **Thanks for the feedback**
>
> “The major issue, which was left as an open question in the end of Section 3, is that when and where to use fortified layers. The authors discussed this issue, but did not solve this issue. Nevertheless, I do believe solving this issue requires a sequence of papers. Overall the paper reads very well, but there are a number of minor places to be improved.“
>
> We thank the reviewer for the positive and constructive feedback.
>
> We also like to point out that we’ve conducted new experiments to help to demonstrate that our method isn’t benefiting from obfuscated gradients and additionally we ran PGD attacks with many more iterations (200) on CIFAR-10 (see the response to reviewer 4).

---

### Official Review · AnonReviewer1 · 2018-11-04
**Improving the robustness of deep Networks by modeling the manifold of hidden representations is original, efficient and well motivated**

**Rating:** 9
**Confidence:** 4

**Review:**

The method works by substituting a hidden layer with a denoised version.
Not only it enable to provide more robust classification results, but also to sense and suggest to the analyst or system when the original example is either adversarial or from a significantly different distribution.
Improvements in adversarial robustness on three datasets are significant.

Bibliography is good, the text is clear, with interesting and complete experimentations.

---

> ### Author Response · Authors · 2018-11-15
> **Thanks for the feedback**
>
> “The method works by substituting a hidden layer with a denoised version.
> Not only it enable to provide more robust classification results, but also to sense and suggest to the analyst or system when the original example is either adversarial or from a significantly different distribution.
> Improvements in adversarial robustness on three datasets are significant.
> Bibliography is good, the text is clear, with interesting and complete experimentations.”
>
> Thank you for your feedback. We have obtained several new results to address concerns related to gradient obfuscation raised by other reviewers and the public comment.

---

### Official Review · AnonReviewer2 · 2018-11-07
**A sensible approach, but needs to justify the experiments more strongly**

**Rating:** 5
**Confidence:** 3

**Review:**

This paper presents an approach of fortifying the neural networks to defend attacks. The major component should be a denoising autoencoder with noise in the hidden layer.

However, from the paper, I am still not convinced why this defends the FGSM attack. From my perspective, a more specifically designed algorithm could attack the network described in the paper as the old way, and what is the insight of defending the attacks, whether this objective function is harder to find to adversarial examples, or have to use more adversarial examples?

Another problem rise from Ian Goodfellow's comment. I am trying not to be biased. So if the author could address his comments properly, I am willing to change the rating.

---

> ### Author Response · Authors · 2018-11-15
> **Thank you for your feedback.**
>
> “This paper presents an approach of fortifying the neural networks to defend attacks. The major component should be a denoising autoencoder with noise in the hidden layer.
> However, from the paper, I am still not convinced why this defends the FGSM attack. From my perspective, a more specifically designed algorithm could attack the network described in the paper as the old way”
>
> In our experiments (except where we explicitly test against a special BPDA) we backpropagate errors through the autoencoders, such that the the autoencoders are not hidden from the attacker.  Indeed we found that skipping the autoencoders when running the attacks makes them significantly weaker, but in our main experiments we backpropagate through the autoencoders and allow the attacker to use this information.
>
> “and what is the insight of defending the attacks, whether this objective function is harder to find to adversarial examples, or have to use more adversarial examples?”
>
> Our main claim is that using a model which can perform reconstruction can map points off of the data manifold back onto the manifold.  For example, we can imagine that unusual noise patterns would not appear in the reconstructions.  These off-manifold points are not necessarily adversarial examples and not all adversarial examples are from off of the manifold (Gilmer 2018).  However, our claim is only that *some* of the adversarial examples are off of the manifold, and thus when we use adversarial training, it is more effective and efficient when we have the autoencoders in the network, as it reduces the space that we need to search over.
>
> Evidence that some adversarial examples are off of the manifold (at least for an undefended network) is in our paper in figure 1.  Some additional qualitative evidence supporting this claim is provided by another submission.  Figure 2 and Figure 3 of the “Robustness May be at Odds with Accuracy” paper (https://openreview.net/pdf?id=SyxAb30cY7), show the perturbations for a defended model appear to be somewhat unrealistic (although much less so then for an undefended model).
>
> “Another problem rise from Ian Goodfellow's comment. I am trying not to be biased. So if the author could address his comments properly, I am willing to change the rating.”
>
> We strongly believe that it is essential to show that the improvements do not result from gradient obfuscation as well as to demonstrate improvements against strong attacks (such as PGD) on CIFAR-10. We have thus run additional experiments demonstrating effectiveness of the method on CIFAR-10, on a CNN as well as a ResNet architecture. We ran validation experiments to confirm that our method does not simply operate by obfuscating gradients.

---

> ### Author Response · Authors · 2018-12-02
> **New Results Summary**
>
> Hello,
>
> We've updated the paper with new results on the PGD attack with many more iterations, architectures (including large architectures like wideresnet), and setups (especially see Tables 2 and 3).  This directly addresses the over-reliance on FGSM as an attack, which was the focus of Ian Goodfellow's comment.

---

### Official Review · AnonReviewer4 · 2018-11-10
**Empirical results are not sufficient to demonstrate the strength of the proposed defense**

**Rating:** 4
**Confidence:** 5

**Review:**

This paper proposes a new defense to adversarial examples based on the 'fortification' of hidden layers using a denoising autoencoder. While building models that are robust to adversarial examples is an important and relevant research problem, I am not convinced by the evaluation of the defense.  Specific comments:

- The authors mostly evaluate their defense using FGSM (particularly on CIFAR). To truly establish the merit of a new defense, the authors must benchmark against state-of-the-art defenses such as PGD. It also seems like the epsilon values used for the PGD attacks are fairly small. The authors should report accuracies to a range of epsilon values for the PGD attack, as is standard.

- When the authors attack their models using PGD/FGSM, is this only on the classification loss or does this also include the denoising terms? Similar defenses which use denoisers have been broken once you run PGD on the full model [1].

- I do not really understand the motivation behind using an autoencoder here. Firstly, it is not clear that adversarial examples lie off the data manifold - they could form a very small set on the data manifold and thereby not affect standard generalization. Secondly, have the authors tried a simple regularization loss based on the error between hidden layer representations to a natural examples and the corresponding adversarial example? I think the authors must motivate the use of denoising autoencoders here by comparing to such a simple baseline.

General comment: The results hard to parse given the arrangement of figures and tables. Also, which approach are the authors denoting as ‘baseline adv. Train’ in the tables?

Overall I feel like building defenses to adversarial examples is a challenging problem and the empirical investigation in this paper is not sufficient to illustrate any real progress on this front.

[1] Athalye, A., & Carlini, N. (2018). On the Robustness of the CVPR 2018 White-Box Adversarial Example Defenses. arXiv preprint arXiv:1804.03286.

---

> ### Author Response · Authors · 2018-11-15
> **Thanks - Response to Comments on Experiments and Gradient Obfuscation**
>
> Thank you for your feedback.  We strongly agree that it is absolutely essential to show that the improvements are not a result of gradient obfuscation.
>
> “The authors mostly evaluate their defense using FGSM (particularly on CIFAR). To truly establish the merit of a new defense, the authors must benchmark against state-of-the-art defenses such as PGD. ”
>
> We added new results with PGD on CIFAR-10 with many more PGD-steps for evaluation.  We evaluated a convolutional network on CIFAR-10 with 4 convolutional layers followed by a single fully-connected layer.  We trained fortified networks, where we added an autoencoder following each hidden layer.  We also added a baseline “Extra Layers” where we trained with the layers added to match the capacity of Fortified Networks (same number of parameters).
>
> # steps | Baseline | Baseline w/ extra layers | Fortified Networks
>     7 steps | 33.0 | 34.2 | 45.0
>   50 steps | 31.6 | 32.5 | 42.1
> 200 steps | 31.4 | 32.2 | 41.5
>
> Even when running PGD for 200 steps, we found large and consistent advantages for fortified networks, which are not primarily attributable to adding additional layers.
>
> “It also seems like the epsilon values used for the PGD attacks are fairly small. The authors should report accuracies to a range of epsilon values for the PGD attack, as is standard.”
>
> This is a great point and we performed an additional experiment using the same convolutional neural network discussed above.  Using 7 and 100 steps of PGD, we attacked our fortified nets model with varying epsilons:
>
> PGD, 100 steps
> Epsilon | Baseline with extra layers | Fortified Networks
> 0.03 | 35.3 | 39.2
> 0.04 | 24.8 | 28.0
> 0.06 | 14.3 | 15.6
> 0.08 | 12.0 | 13.0
>   0.1 | 11.7 | 12.9
>   0.2 | 10.2 | 11.3
>   0.3 |   8.4 | 9.6
>
> “When the authors attack their models using PGD/FGSM, is this only on the classification loss or does this also include the denoising terms? Similar defenses which use denoisers have been broken once you run PGD on the full model [1].”
>
> We run our attacks on the full model, end-to-end, including the autoencoders. This is a major difference from [1], and that change is what broke the paper you referenced.  The paper that you referenced did not perform adversarial training on the main part of the network, and only trained the autoencoder, keeping the classifier network itself fixed.
>
> We also conducted a new experiment with BPDA (Athalye 2018), where we consider skipping the autoencoders in the backward pass (i.e. using the identity function to compute the gradients) as well as running the forward and backward pass of the network with no noise injected.
>
> We also ran some new experiments for this using eps=0.03 and 100 steps of PGD using the same CNN architecture discussed earlier.
>
> 33.4 (baseline, normal attack)
> 40.1 (Fortified Networks, normal attack)
> 38.2 (Fortified Networks, no noise during attack)
> 67.1 (Fortified Networks, skip DAE during attack, BPDA)
>
> This is strong evidence that skipping the autoencoders while generating the attacks significantly weakens them, but turning the noise off slightly strengthens the attack, but it is still much stronger as a defense than the baseline adversarially trained model with the same number of parameters and capacity.
>
> “Secondly, have the authors tried a simple regularization loss based on the error between hidden layer representations to a natural examples and the corresponding adversarial example? I think the authors must motivate the use of denoising autoencoders here by comparing to such a simple baseline”
>
> Yes, we conducted new experiments to directly address this issue (Adversarial Logit Pairing is a special case of the regularizer that you describe), but we also note that our method provides improvements even when we don’t use the L_adv loss comparing the adversarial input’s hidden states to the clean input’s h.  This is with the same CNN architecture discussed earlier.
>
> PGD, 7 iterations:
> 43.3 (Fortified Networks)
> 38.1 (Adv. training baseline)
> 34.2 (Penalty between layers)
>
> PGD, 100 iterations:
> 39.2 (Fortified Networks)
> 35.3 (Adv. training baseline)
> 32.2 (Penalty between layers)
>
> We found that this penalty between the hidden states, where we attracted the hidden states in the network on adversarial inputs to the hidden states of the network on clean states (applied at every layer), hurts robustness somewhat, but it may be possible that such an approach depends on exactly how it’s used.  We also add that unlike adversarial logit pairing, our improvements hold up after running PGD for a large number of iterations, whereas the benefits from adversarial logit pairing almost entirely disappear.
>
> “Also, which approach are the authors denoting as ‘baseline adv. Train’ in the tables?“
>
> This refers to the PGD training of Madry 2017.

---

> > ### Public Comment · (anonymous) · 2018-11-17
> > **SPSA?**
> >
> > Can you try out SPSA and confirm your results? The public discussion on the link below shows that PGD with large iterations cannot actually detect masked gradients fully, but SPSA sort of pretty much cancels all their gains from their method!
> >
> > See discussion on this page.
> > https://openreview.net/forum?id=Bylj6oC5K7&noteId=H1leI9Iah7

---

> > ### Public Comment · (anonymous) · 2018-11-18
> > **Why is your baseline weaker than Madry et al.?**
> >
> > You claim that PGD adversarial training as a baseline gives a robustness of 35% at eps=0.03. However, to the best of my knowledge, no prior paper has reduced the accuracy below 44%.
> >
> > Can you account for this difference? Are you able to lower the accuracy the Madry et al. defense to 38%?
> >
> > While a gap of ~6% might not typically be important, you are only claiming a gain of about 5%. So in this case, it's absolutely critical that we can be sure it's not just that you have a weak baseline you're comparing against.

---

> > > ### Author Response · Authors · 2018-11-18
> > > **Reason**
> > >
> > > Thanks, the reason is that the baseline is a 4-layer CNN and not a resnet.  When we run with the resnet our results are about the same as Madry, but our goal in the rebuttal has been to get the results with as many types of attacks/setups as possible to ensure that the improvements are a not result of gradient masking.
> > >
> > > We can add more experiments with ResNets as well.

---

> > > > ### Public Comment · (anonymous) · 2018-11-19
> > > > **Makes sense**
> > > >
> > > > Thank you, that makes sense.
> > > >
> > > > I agree running a resnet would be very important. Madry et al. state that one of the reasons their defense works is that you have to have large network capacity.

---

> > ### Comment · AnonReviewer4 · 2018-11-19
> > **Response to Experiments**
> >
> > I am still not convinced by the empirical evaluation performed by the authors. My concerns are:
> >
> > 1. The proposed method is not an alternative to adversarial training, but instead augments it with an additional objective from the denoising autoencoder. The authors are also claiming only ~5% improvement over the baseline. One might argue that the benefits of the proposed approach over adversarial training are marginal. Even if we assume that the 5% is significant, it is not clear how accurate the baseline evaluation is. I agree with one of the anonymous comments in this regard. The authors use a non-standard model, and their PGD baseline is quite a bit lower than the state-of-the-art. I would really like to see the results on a state-of-the-art model to be convinced that the benefit is not just an artifact of a weak baseline.
> >
> > 2. If I correctly understand the new results posted by the authors, their model obtains ~10-13% accuracy against an Linf adversary of eps>0.1 on CIFAR-10. It has been shown that an eps~0.125 is already too large - one can perturb the image to actually be from another class (also shown in the ICLR submission that the authors linked - “Robustness may be at odds with accuracy” https://openreview.net/forum?id=SyxAb30cY7). I do not understand how the fortified model can get an accuracy > 0% for such large epsilons, which are probably impossible to be robust to. Have the authors checked what the adversarial examples look like for these large eps? What about trying a nearest neighbor attack from the test set? Seeing a non-zero robust accuracy to such large epsilons makes me doubt the correctness of the attack setup within the experimental evaluation.
> >
> > 3. The proposed defense seems to use random noise (as part of the denoising stage). Have the authors tried multiple gradient queries per PGD step?
> >
> > 4. I would also like to see the standard (non-robust) accuracies of the models (specifically the  baseline model, baseline with extra layers and fortified networks) to make sure that the 5% gain in robustness is not an artifact of larger expressivity of the proposed model.

---

> > > ### Public Comment · (anonymous) · 2018-11-24
> > > **Comment 2. does not seem fully accurate**
> > >
> > > https://arxiv.org/pdf/1706.06083.pdf
> > >
> > > The original paper linked above claims an accuracy of ~10% at eps=30/255 (eps~0.118). I do not think the argument that "adversarial examples exist => PGD will find it" is accurate. In fact, https://arxiv.org/pdf/1706.06083.pdf  has a disclaimer that suggests such points may well be present and PGD may well be unable to find them.

---

> > > ### Author Response · Authors · 2018-11-26
> > > **Thanks for the Feedback - Response**
> > >
> > > “1. The proposed method is not an alternative to adversarial training, but instead augments it with an additional objective from the denoising autoencoder. The authors are also claiming only ~5% improvement over the baseline. One might argue that the benefits of the proposed approach over adversarial training are marginal. Even if we assume that the 5% is significant, it is not clear how accurate the baseline evaluation is. I agree with one of the anonymous comments in this regard. The authors use a non-standard model, and their PGD baseline is quite a bit lower than the state-of-the-art. I would really like to see the results on a state-of-the-art model to be convinced that the benefit is not just an artifact of a weak baseline.”
> > >
> > > We conducted experiments using two much stronger models: PreActResNet18 and WideResNet28-10.  All experiments ran for 200 epochs.
> > >
> > > PreActResNet18
> > > Baseline: 37.87 (20 step PGD), 84.93% (clean test accuracy)
> > > Fortified Networks: 39.2% (20 step PGD), 84.84% (clean test accuracy)
> > >
> > > WideResNet28-10:
> > > Baseline: 43.28% (20 step PGD), 87.42% (clean test accuracy)
> > > Fortified Networks: 44.06% (20 step PGD), (87.40% clean test accuracy)
> > >
> > > “2. If I correctly understand the new results posted by the authors, their model obtains ~10-13% accuracy against an Linf adversary of eps>0.1 on CIFAR-10. It has been shown that an eps~0.125 is already too large - one can perturb the image to actually be from another class (also shown in the ICLR submission that the authors linked - “Robustness may be at odds with accuracy” https://openreview.net/forum?id=SyxAb30cY7). I do not understand how the fortified model can get an accuracy > 0% for such large epsilons, which are probably impossible to be robust to. Have the authors checked what the adversarial examples look like for these large eps? What about trying a nearest neighbor attack from the test set? Seeing a non-zero robust accuracy to such large epsilons makes me doubt the correctness of the attack setup within the experimental evaluation.”
> > >
> > > Our robustness with an epsilon of 0.3 is very similar to what’s reported in (Madry 2018), especially Figure 6c:
> > >
> > > https://openreview.net/pdf?id=SyxZJn05YX
> > >
> > >
> > > One possibility is that for some examples, it is possible to find a real example with a different class within an epsilon ball of size 0.3 - but there is a small fraction of examples where this isn’t possible.
> > >
> > > “3. The proposed defense seems to use random noise (as part of the denoising stage). Have the authors tried multiple gradient queries per PGD step? “
> > >
> > > We conducted a very similar experiment to this where we ran both the forward and the backward pass without any injected noise and we showed that Fortified Networks retained a significant improvement over the baseline.

---

> ### Author Response · Authors · 2018-11-15
> **Motivation for Fortified Networks**
>
> “ I do not really understand the motivation behind using an autoencoder here. Firstly, it is not clear that adversarial examples lie off the data manifold - they could form a very small set on the data manifold and thereby not affect standard generalization.”
>
> Our main claim is that using a model which can perform reconstruction can map points off of the data manifold back onto the manifold.  For example, we can imagine that unusual noise patterns would not appear in the reconstructions.  These off-manifold points are not necessarily adversarial examples and not all adversarial examples are from off of the manifold (Gilmer 2018).  However, our claim is only that *some* of the adversarial examples are off of the manifold, and thus when we use adversarial training, it is more effective and efficient when we have the autoencoders in the network, as it reduces the space that we need to search over.
>
> Evidence that some adversarial examples are off of the manifold (at least for an undefended network) is in our paper in figure 1.  Some additional qualitative evidence supporting this claim is provided by another submission.  Figure 2 and Figure 3 of the “Robustness May be at Odds with Accuracy” paper (https://openreview.net/pdf?id=SyxAb30cY7), show the perturbations for a defended model appear to be somewhat unrealistic (although much less so then for an undefended model).

---

### Public Comment · (anonymous) · 2018-11-06
**Experimental Evaluation Not Convincing**

I like the writing, but I have some core problems with the experimental evaluation.

Some questions:
1. Why is CIFAR only evaluated against FGSM? Shouldn't you at least try PGD on CIFAR-10? Why not try out PGD/CW on CIFAR-10?  It is not obvious that the method will scale to complex datasets such as CIFAR-10 (leave alone Imagenet).

2. Why not try out NES/SPSA/ElasticNet attacks as evidence against gradient-masking?

3. Blackbox accuracy seems to be slightly worse than white-box.  Is this a sign that there is some gradient masking going on?

4. For how many iterations was PGD run? I think this information is critical. How many random restarts? There is some recent work (https://arxiv.org/abs/1810.12042) that indicates large number of restarts/iteration steps might be necessary for a meaningful evaluation

5. Why not baseline against adversarial logit pairing? (investigations by third parties have shown that while ALP does not help as much as claimed with Imagenet, it does help with CIFAR and MNIST).

The evidence against gradient masking given in the paper is also presented by ALP (https://arxiv.org/abs/1803.06373). But, this paper (https://arxiv.org/abs/1807.10272) shows that these signs may very well be present in defenses that rely on gradient obfuscation.

Overall, there are no theoretical guarantees and I am not convinced that there are actually any gains compared to ALP/other SOTA defenses... especially with the fact that the possibility of gradient obfuscation has not been fully explored, and that most experiments are limited to MNIST!

---

> ### Author Response · Authors · 2018-11-17
> **Thanks for your feedback**
>
> We thank the commenter for their valuable feedback and suggestions for more thorough experimentation. We have run many of the suggested tests to address the question of gradient obfuscation, which was also raised by others.
>
> “Why is CIFAR only evaluated against FGSM? Shouldn't you at least try PGD on CIFAR-10? Why not try out PGD/CW on CIFAR-10?  It is not obvious that the method will scale to complex datasets such as CIFAR-10 (leave alone Imagenet).”
>
> We have added new results with PGD on CIFAR-10 with many more iterations at evaluation time.
>
> # steps | Baseline | Baseline w/ extra layers | Fortified Networks
>     7 steps | 33.0 | 34.2 | 45.0
>   50 steps | 31.6 | 32.5 | 42.1
> 200 steps | 31.4 | 32.2 | 41.5
>
> “For how many iterations was PGD run? I think this information is critical. How many random restarts? There is some recent work (https://arxiv.org/abs/1810.12042) that indicates large number of restarts/iteration steps might be necessary for a meaningful evaluation”
>
> We have added new results with PGD run for many iterations (up to 200), and with several restarts (up to 50), as well as for different epsilon values (0.03 to 0.3). Our model outperforms baseline models in all cases, demonstrating effectiveness of the method even under these more difficult conditions.
>
> “Why not baseline against adversarial logit pairing? (investigations by third parties have shown that while ALP does not help as much as claimed with Imagenet, it does help with CIFAR and MNIST).”
>
> We ran ALP-like experiments, wherein we added an adversarial loss on the hidden layers instead of adding fortified layers . We could not achieve competitive performance with this system. In fact, it did not perform better than just an adversarially trained baseline system, however, we have not exhaustively explored this approach.

---

> > ### Public Comment · (anonymous) · 2018-11-20
> > **This paper seems to indicate ALP results in robustness, comparable to your approach**
> >
> > https://arxiv.org/pdf/1810.12042.pdf
> >
> > However, they may be complementary and can be combined?

---

### Public Comment · ~Ian_Goodfellow1 · 2018-11-06
**FGSM is not a strong attack**

I'm not trying to weigh in on whether or not the paper should be accepted and I haven't read the paper; I'm just trying to provide the reviewers with good information on how to interpret FGSM experiments. I'm commenting because a colleague told me that this information would be relevant to reviewing this paper.

I developed the FGSM attack, and I'd like to comment that it's not intended to be a strong attack.

The FGSM was mostly intended to be used for a scientific experiment to show that linear information is sufficient to break undefended neural nets. It's not meant to be a strong attack.

Until a few years ago, FGSM was also a good "unit test" to see if a defense was strong. By now, I personally don't even use FGSM as a unit test anymore. Performance on FGSM does not correlate well with performance on the strongest attacks.

It's fine if you want to use FGSM as a unit test but success on FGSM shouldn't be regarded as strong evidence that a defense works in a particular threat model. The reviewers should check what specific claims are made in the paper and if there are claims of a strong defense these claims should be supported by something other than FGSM.

---

### Public Comment · ~Ian_Goodfellow1 · 2018-11-06
**Thermometer coding does not improve adversarial robustness**

I haven't read this paper but a colleague told me that it quotes the accuracy numbers from the original Buckman et al paper and uses them as a point of comparison. I'm a co-author of thermometer coding, and I'm here to say it's important to understand that a new attack, BPDA, was able to break the model from our paper: https://arxiv.org/abs/1802.00420

In our own follow-up experiments, we found that if we retrain using BPDA for adversarial training, models that use thermometer coding perform about the same as models that use real numbers for input. Thus it's probably best to just use adversarial training as the baseline.

---

> ### Author Response · Authors · 2018-11-15
> **Thanks**
>
> We have removed the thermometer coding reference from the results table and we have also run new experiments to attack fortified networks using BPDA.

---

### Public Comment · (anonymous) · 2018-11-08
**Is there proof for the main claim?**

Is there any proof that using an autoencoder maps the data back in the manifold? Especially against adversarial perturbations?

Have the authors tried their method with networks that include residual connections? It will be interesting to verify that mapping back to the manifold indeed works with such connections that can amplify perturbations through the skip connections.

---

> ### Author Response · Authors · 2018-11-15
> **Main Claim Clarification**
>
> “Is there any proof that using an autoencoder maps the data back in the manifold? Especially against adversarial perturbations?”
>
> To clarify: our motivation is that the autoencoders map some points from off of the manifold back onto the manifold.  This in turn reduces the potential space of adversarial examples (because most of the space is off-manifold), which then makes adversarial training more efficient. These off-manifold points are not necessarily adversarial examples and not all adversarial examples are off the manifold (Gilmer 2018).  However, our main claim is that some of the adversarial examples are off of the manifold, and thus when we use adversarial training, it is more effective and efficient when we have the autoencoders in the network.
>
> Evidence that some adversarial examples are off of the manifold (at least for an undefended network) is in our paper in figure 1.  Some qualitative evidence supporting this claim is provided by another submission.  Figure 2 and Figure 3 of the “Robustness May be at Odds with Accuracy” paper (https://openreview.net/pdf?id=SyxAb30cY7), show the perturbations for a defended model appear to be somewhat unrealistic (although much less so then for an undefended model).

---

### Public Comment · (anonymous) · 2018-11-19
**Confusion over new results**

I'm having a hard time interpreting the new results.

- Table 2 CIFAR-10 argues PGD eps=0.03 error of the baseline network is 38.1%.
- Table 8 CIFAR-10 argues PGD eps=0.03 error of the baseline network is 33.0% or 31.4% for 7 or 200 (respectively) iterations of gradient descent. Why is this different? How many iterations did you use in Table 2?

- Table 7 argues 100 iterations of PGD at eps=0.03 has an error rate of 35.3% on "basline with extra layers"
- Table 8 argues 50/200 iterations of PGD at eps=0.03 has an error rate of 32.5/32.2 (respectively for the same model. Because 50<100<200 I would expect that the 35.3 should be something smaller. Why is this?

Because the improvement gain for fortified networks is relatively small, these ~5% differences add up. Are they due to random initializations? In that case, could we get some margin-of-error results for these tables?

---

> ### Author Response · Authors · 2018-11-26
> **Clarification**
>
> "- Table 2 CIFAR-10 argues PGD eps=0.03 error of the baseline network is 38.1%.
> - Table 8 CIFAR-10 argues PGD eps=0.03 error of the baseline network is 33.0% or 31.4% for 7 or 200 (respectively) iterations of gradient descent. Why is this different? How many iterations did you use in Table 2?
>
> - Table 7 argues 100 iterations of PGD at eps=0.03 has an error rate of 35.3% on "basline with extra layers"
> - Table 8 argues 50/200 iterations of PGD at eps=0.03 has an error rate of 32.5/32.2 (respectively for the same model. Because 50<100<200 I would expect that the 35.3 should be something smaller. Why is this?”
>
> Because Fortified Networks adds capacity to the model, using the network without the fortified layers is a weak baseline.  The discrepancy that you point to results from two different ways of adding activations to the baseline model.  Essentially, the lower result uses the same number of layers with activations as fortified networks, but the higher number has more activations, and in some sense this makes it a higher capacity model.  Nonetheless the paper has been updated with the discrepancy explained (Table 2).

---

### Public Comment · (anonymous) · 2018-11-25
**A closed related paper**

Hi,
I found the idea is very similar to "Defense against Adversarial Attacks Using High-Level Representation Guided Denoiser"
http://openaccess.thecvf.com/content_cvpr_2018/papers_backup/Liao_Defense_Against_Adversarial_CVPR_2018_paper.pdf

Could you please clarify the difference between your work and this paper? Thanks.

---

> ### Author Response · Authors · 2018-11-25
> **Difference Between the Papers**
>
> Hello,
>
> Our method and the "High-Level Representation Guided Denoiser" are very different.  We ran our attacks and evaluate on the full model, end-to-end, including the autoencoders. This is a major difference from [1], and that change is what broke the paper you referenced.  The paper that you referenced did not perform adversarial training on the main part of the network, and only trained the autoencoder, keeping the classifier network itself fixed.
>
> We also conducted an experiment with BPDA (Athalye 2018), where we consider skipping the autoencoders in the backward pass (i.e. using the identity function to compute the gradients) as well as running the forward and backward pass of the network with no noise injected and we produced, and the advantage of fortified networks was preserved.

---

### Author Response · Authors · 2018-11-27
**Rebuttal Summary and Highlights**

We thank all of the reviewers and commenters for their feedback, which has done a great deal to improve the quality of the paper. The main points raised by reviewers and commenters were related to the experimental results, the motivation, gradient obfuscation tests, and related work. All points have been addressed in the revised manuscript, and are summarized in the following.

1.  Stronger Attacks: We strongly agree that the FGSM attack is not a strong attack and to that end we have conducted new experiments against the PGD attack with up to 200 steps as well as a range of epsilons from 0.03 to 0.3 (Table 2).  The improvements from Fortified Networks with 200 steps are similar to the improvements over baseline with 7 steps, and also Fortified Networks improve results over the baseline when using larger epsilons.

2.  Motivation for Fortified Networks: we have clarified that our motivation for fortified networks is that the autoencoders map some points from off of the manifold back onto the manifold.  This in turn reduces the potential space of adversarial examples (because most of the space is off-manifold), which then makes adversarial training more efficient. These off-manifold points are not necessarily adversarial examples and not all adversarial examples are off the manifold (Gilmer 2018).  However, our main claim is that some of the adversarial examples are off of the manifold, and thus when we use adversarial training, it is more effective and efficient when we have the autoencoders in the network.

3.  Gradient Obfuscation and Masking: We strongly agree that it is important to show that the improvements are due to actual improvements in robustness and not merely a degradation in the quality of the gradient signal.  To address this, we have run PGD with a greater number of steps (up to 200).  We have also run some variants of the attack which address issues related to gradient obfuscation (Table 2).  For example we have run with larger epsilons and found that the model is still able to find adversarial attacks.  Additionally we have run the network without noise and with attacks where gradient skips the autoencoder (BPDA), and found that Fortified Networks still improve robustness in both cases.

4.  Baselines and Related Work: We have added new results with PreActResNet18 and WideResNet28-10 on CIFAR-10 (Table 3), which are relatively competitive architectures.  In both cases we found significant improvements using Fortified Networks and intriguingly we saw almost no change in the clean test accuracy.  This is strong evidence that the resulting improvement does not trivially come from added capacity, as was suggested as a possibility by R4.  Additionally we conducted an experiment where we simply added a square loss on the hidden layers (similar to ALP except on all layers) and found that this did not improve results.

---

> ### Comment · AnonReviewer4 · 2018-11-28
> **New experiments show significantly lower gains from fortification**
>
> In the new experiments conducted by the authors on the two ResNet models, the additional benefits of fortification are even less significant (about 1%, close to error margins).
>
> If this degree of robustness was attained by a technique which completely *replaces* adversarial training, I think it would indeed be valuable. But in this paper, the proposed method *augments* adversarial training with additional loss terms, and so one can argue that most of the robustness comes from adversarial training itself and the benefits of the fortified layers are marginal.
>
> Thus, based on the empirical results, I do not think that the contribution of the proposed approach as a defense against adversarial attacks is sufficient.

---

### Meta-Review · Area_Chair1 · 2018-12-15
**The ideas are quite novel and promising, but there is no sufficient justification of claims made**

**Confidence:** 5
**Recommendation:** Reject

**Metareview:**

This paper suggests a method for defending against adversarial examples and out-of-distribution samples via projection onto the data manifold. The paper suggests a new method for detecting when hidden layers are off of the manifold, and uses auto encoders to map them back onto the manifold.

The paper is well-written and the method is novel and interesting. However, most of the reviewers agree that the original robustness evaluations were not sufficient due to restricting the evaluation to using FGSM baseline and comparison with thermometer encoding (which both are known to not be fully effective baselines).

After rebuttal, Reviewer 4 points out that the method offers very little robustness over adversarial training alone, even though it is combined with adversarial training, which suggests that the method itself provides very little robustness.